# Learning Disentangled Stain and Structural Representations for Semi-Supervised Histopathology Segmentation

**Ha-Hieu Pham**[1,3,7*]                          PHHIEU22@CLC.FITUS.EDU.VN
**Nguyen Lan Vi Vu**[2,3*]                        VI.VUVIVU2203@HCMUT.EDU.VN
**Thanh-Huy Nguyen**[4]                           THANHHUN@ANDREW.CMU.EDU
**Ulas Bagci**[5]                                 ULAS.BAGCI@NORTHWESTERN.EDU
**Min Xu**[4]                                     MXU1@CS.CMU.EDU
**Trung-Nghia Le**[1,3]                           LTNGHIA@FIT.HCMUS.EDU.VN
**Huy-Hieu Pham**[6,7,†]                          HIEU.PH@VINUNI.EDU.VN

[1] *University of Science, VNU-HCM, Ho Chi Minh City, Vietnam*

[2] *Ho Chi Minh University of Technology, Ho Chi Minh City, Vietnam*

[3] *Vietnam National University, Ho Chi Minh City, Vietnam*

[4] *Carnegie Mellon University, Pittsburgh, PA, USA*

[5] *Northwestern University, Chicago, IL, USA*

[6] *College of Engineering & Computer Science, VinUniversity, Ha Noi City, Vietnam*

[7] *VinUni-Illinois Smart Health Center, VinUniversity, Ha Noi City, Vietnam*

[*] *These authors contributed equally to this work.*

[†] *Corresponding author: hieu.ph@vinuni.edu.vn (Huy-Hieu Pham)*

## Abstract

Accurate gland segmentation in histopathology images is essential for cancer diagnosis and prognosis. However, significant variability in Hematoxylin and Eosin (H&E) staining and tissue morphology, combined with limited annotated data, poses major challenges for automated segmentation. To address this, we propose Color-Structure Dual-Student (CSDS), a novel semi-supervised segmentation framework designed to learn disentangled representations of stain appearance and tissue structure. CSDS comprises two specialized student networks: one trained on stain-augmented inputs to model chromatic variation, and the other on structure-augmented inputs to capture morphological cues. A shared teacher network, updated via Exponential Moving Average (EMA), supervises both students through pseudo-labels. To further improve label reliability, we introduce stain-aware and structure-aware uncertainty estimation modules that adaptively modulate the contribution of each student during training. Experiments on the GlaS and CRAG datasets show that CSDS achieves state-of-the-art performance in low-label settings, with Dice score improvements of up to 1.2% on GlaS and 0.7% on CRAG at 5% labeled data, and 0.7% and 1.4% at 10%. Our code and pre-trained models are available at https://github.com/hieuphamha19/CSDS.

**Keywords:** Histopathology Imaging, Semi-Supervised Learning, Semantic Segmentation.

## 1 Introduction

Histopathological image analysis is of paramount importance in cancer diagnosis and prognosis, especially in tasks such as gland segmentation for colorectal cancer. Images are usually stained with Hematoxylin and Eosin (H&E). This staining can create significant color differences due to variations in staining methods, slide preparation, and scanning equipment used. The differences in color and structure of tissue images create considerable obstacles

for automated segmentation systems, especially when there is a lack of labeled data. In medical imaging, however, obtaining such labels is expensive, time-consuming, and subject to inter-observer variability.

Semi-supervised learning (SSL) offers an effective strategy to improve segmentation performance by leveraging a small amount of labeled data alongside a larger pool of unlabeled samples. However, existing SSL methods often overlook the unique visual complexity of histopathological images, which arises from two entangled yet semantically distinct sources: staining variations and tissue morphology. In particular, Hematoxylin and Eosin (H&E) staining introduces substantial inter-sample color variation due to inconsistencies in staining protocols and scanning devices, while structural variations reflect biologically meaningful changes in tissue architecture, including malignant progression. Prior works in semi-supervised segmentation, such as pseudo-labeling, consistency regularization Sohn et al. (2020); Le et al. (2025), and teacher-student frameworks Tarvainen and Valpola (2017); Yu et al. (2019); Pham et al. (2025), generally process the image holistically and fail to disentangle stain and structure cues. This lack of targeted representation learning can limit their effectiveness in histopathology, where isolating stain-specific and structure-specific information is crucial for robust and interpretable segmentation.

To overcome the limitations of existing semi-supervised segmentation approaches, we propose in this work a novel framework termed **Color-Structure Dual-Student (CSDS)**. The key innovation of CSDS lies in its explicit decoupling of color and structural information through the use of two specialized student networks. One student is trained with color-augmented images to effectively model chromatic variations, while the other is trained with geometrically transformed images to emphasize structural cues. A shared teacher model, updated using an Exponential Moving Average (EMA) of the student weights, then serves as a pseudo-label generator and provides supervision in an uncertainty-aware manner.

To further enhance pseudo-label reliability, we introduce color-aware and structure-aware uncertainty estimation strategies. These mechanisms leverage prediction entropy and domain-specific priors to adaptively weight the contributions of each student during training, thereby improving the quality and robustness of the supervision signal. We validate the effectiveness of CSDS through comprehensive experiments conducted on two histopathological image segmentation benchmarks, GlaS Sirinukunwattana et al. (2017) and CRAG Graham et al. (2019), under low-label regimes. The results demonstrate that CSDS consistently outperforms existing state-of-the-art methods, establishing a new benchmark for semi-supervised medical image segmentation.

## 2 Methodology

### 2.1 Preliminary

**Problem Settings.** Following standard semi-supervised segmentation settings, we denote the labeled set as $\mathcal{D}_l = \{(x_i^l, y_i^l)\}_{i=1}^N$, and a larger unlabeled set as $\mathcal{D}_u = \{(x_k^u)\}_{k=1}^M$ ($N$: labeled samples, $M$: unlabeled samples, $M \gg N$). Here, $x_i^l$ and $x_k^u$ are input images from the labeled and unlabeled sets, respectively, and $y_i^l$ is the corresponding one-hot groundtruth for labeled sample $x_i^l$. This setup formalizes the standard semi-supervised segmentation problem under a limited-label regime.

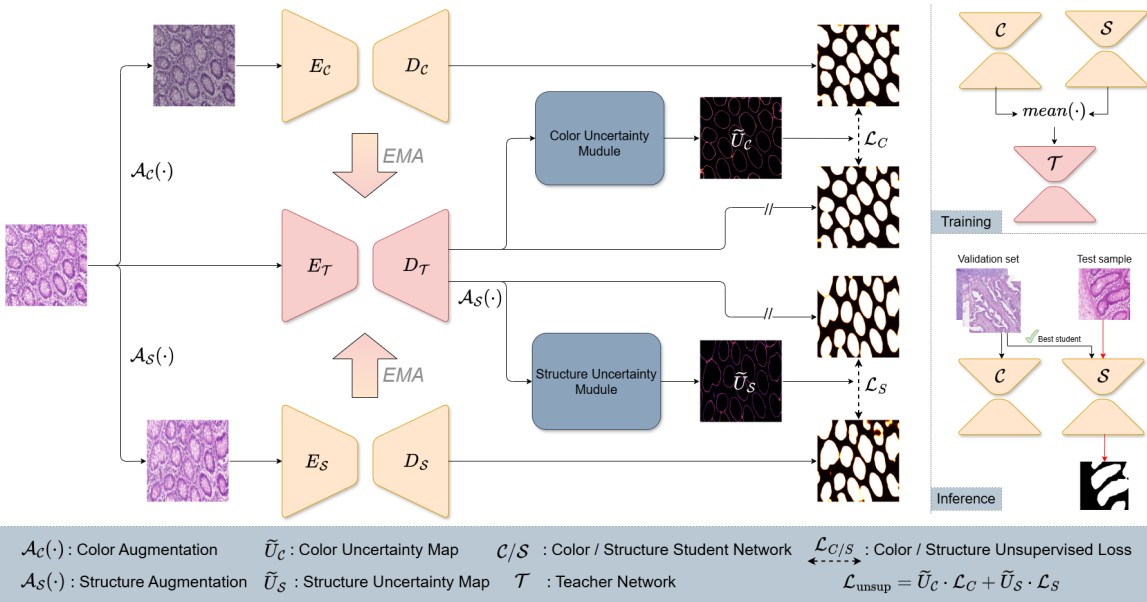

Figure 1: Overview of our Color-Structure Dual-Student (CSDS) framework with dual students for color and structure, and a shared EMA-updated teacher. Output uncertainty maps are used to adaptively weight pseudo-labels during training.

**Overall Pipeline.** We propose a Color-Structure Dual-Student (CSDS) framework, an extension of the Mean Teacher paradigm Tarvainen and Valpola (2017). Our framework is built upon the intuition that color and structure represent two complementary sources of variation in histological data: while color distributions often shift across staining protocols, structural patterns exhibit large inter-sample variability due to tissue deformation and cancer heterogeneity. CSDS comprises two student networks with shared architecture: the color student, trained on inputs augmented by histogram matching and color jittering to capture both in-distribution and out-of-distribution color shifts; and the structure student, optimized on elastically deformed images to model structural variability. A shared teacher network is updated via Exponential Moving Average (EMA) of the mean weights from both students. In Section 2.2 we introduce two complementary uncertainty estimation modules: Color uncertainty and structure uncertainty, which quantify prediction confidence via entropy in color-ambiguous and structurally-uncertain regions, respectively. These uncertainty maps are subsequently used to modulate the teacher's supervision signals during the pseudo-labeling process. The overall pipeline of our method is illustrated in Figure 1.

## 2.2 Uncertainty Estimation for Color-Structure Uncertainty Module

In this section, we present the detailed design of the Color Uncertainty and Structure Uncertainty modules. Uncertainty map is initially computed from the teacher's predictions using Shannon entropy Shannon (1948) and is further refined by incorporating visual features extracted from input images, guided by color and structural indicators Zou et al. (2023).

### 2.2.1 ENTROPY-BASED UNCERTAINTY ESTIMATION

Given the teacher's output logits $\mathbf{z} \in \mathbb{R}^{C \times H \times W}$, we compute per-pixel softmax probabilities:

$$\mathbf{p}(x) = \text{softmax}(\mathbf{z}(x)), \tag{1}$$

where $z(x)$ denotes the logit at pixel $x$. Uncertainty is then estimated as:

$$U(x) = -\sum_{c=1}^{C} p_c(x) \log(p_c(x) + \varepsilon), \tag{2}$$

where $\varepsilon$ is a small constant added for numerical stability. This produces a dense uncertainty map that reflects the teacher's predictive confidence.

### 2.2.2 COLOR-AWARE UNCERTAINTY MODULATION

Within the domain of pathology imaging, local color diversity often reflects meaningful semantic variation Xia et al. (2024). Given the input RGB image $I \in \mathbb{R}^{3 \times H \times W}$, we estimate chromatic complexity by computing the per-pixel inter-channel variance:

$$\sigma^2(x) = \text{Var}_{c \in \{R,G,B\}}[I_c(x)]. \tag{3}$$

To enforce spatial coherence, the variance map is smoothed using a Gaussian filter $\mathcal{G}_{3\times3}$, or approximated via average pooling for computational efficiency:

$$V(x) = \mathcal{G}_{3\times3}(\sigma^2(x)), \quad \hat{V}(x) = \frac{V(x)}{\max_{x \in \Omega} V(x) + \varepsilon}. \tag{4}$$

A binary mask highlights regions with high color diversity:

$$M_{\mathcal{C}}(x) = \mathbb{I}[\hat{V}(x) > \tau_{\text{color}}]. \tag{5}$$

We then modulate the base uncertainty map $U(x)$ by amplifying values in chromatically distinctive regions:

$$\widetilde{U}_{\mathcal{C}}(x) = U(x) \cdot (1 + \lambda_{\mathcal{C}} \cdot M_{\mathcal{C}}(x)), \tag{6}$$

where $\lambda_{\mathcal{C}} \in [0, 1]$ controls the degree of modulation. This adjustment selectively increases uncertainty in visually rich areas where semantic ambiguity may be elevated.

### 2.2.3 STRUCTURE-AWARE UNCERTAINTY MODULATION

Uncertainty tends to rise near object boundaries and regions of strong visual transition, where label ambiguity is more likely Yang et al. (2025). To model this, we compute approximate spatial gradients using finite differences:

$$G_h(x) = |I(x+1, y) - I(x, y)|, \quad G_v(x) = |I(x, y+1) - I(x, y)|. \tag{7}$$

An edge magnitude map is formed and normalized:

$$E(x) = \frac{1}{3}(G_h(x) + G_v(x)), \quad \hat{E}(x) = \frac{E(x)}{\max_{x \in \Omega} E(x) + \varepsilon}. \tag{8}$$

We then identify structurally salient regions:

$$M_{\mathcal{S}}(x) = \mathbb{I}[\hat{E}(x) > \tau_{\text{structure}}]. \tag{9}$$

Finally, the uncertainty map is further refined by emphasizing these structure-rich areas:

$$\widetilde{U}_{\mathcal{S}}(x) = \widetilde{U}_{\mathcal{C}}(x) \cdot (1 + \lambda_{\mathcal{S}} \cdot M_{\mathcal{S}}(x)), \tag{10}$$

where $\lambda_{\mathcal{S}} \in [0, 1]$ controls the strength of modulation. The final uncertainty map $\widetilde{U}_{\mathcal{S}}$ integrates both chromatic and structural information, yielding a perception-aware enhancement that emphasizes regions of higher epistemic risk.

## 2.3 Training Procedure

We denote the teacher network as $\mathcal{T}$, the color student as $\mathcal{C}$, and the structure student as $\mathcal{S}$. In the supervised phase, the loss is computed as:

$$\mathcal{L}_{\text{sup}} = \mathcal{L}_{\text{CE+Dice}}(\mathcal{C}(x^l), y^l) + \mathcal{L}_{\text{CE+Dice}}(\mathcal{S}(x^l), y^l), \tag{11}$$

where $\mathcal{L}_{\text{CE+Dice}}$ denotes the average of the cross-entropy loss and Dice loss. In the unsupervised phase, for each unlabeled sample $x^u \in \mathcal{D}_u$, we define two augmentation pipelines: $\mathcal{A}_{\mathcal{C}} \in$ {color jittering, histogram matching} for color perturbation and $\mathcal{A}_{\mathcal{S}} \in$ {elastic transformation} for structural deformation. Pseudo-labels are generated by the teacher $\mathcal{T}$, and consistency is enforced through uncertainty-weighted losses:

$$\begin{aligned}
\mathcal{L}_{\text{unsup}} = {} & \widetilde{U}_{\mathcal{C}}(x^u) \cdot \mathcal{L}_{\text{CE+Dice}}(\mathcal{C}(\mathcal{A}_{\mathcal{C}}(x^u)), \mathcal{T}(x^u)) \\
& + \widetilde{U}_{\mathcal{S}}(x^u) \cdot \mathcal{L}_{\text{CE+Dice}}(\mathcal{S}(\mathcal{A}_{\mathcal{S}}(x^u)), \mathcal{T}(\mathcal{A}_{\mathcal{S}}(x^u)))
\end{aligned} \tag{12}$$

Finally, the total training loss combines both objectives:

$$\mathcal{L}_{\text{total}} = \mathcal{L}_{\text{sup}} + \lambda_{\text{unsup}} \mathcal{L}_{\text{unsup}}, \tag{13}$$

where $\lambda_{\text{unsup}}$ is a weighting factor that controls the unsupervised loss contribution.

## 3 Experiments

### 3.1 Datasets

We evaluated our proposed method on two publicly available histopathological benchmarks. **GlaS** Sirinukunwattana et al. (2017) is a gland segmentation dataset for colorectal adenocarcinoma across various cancer stages, containing 165 images (37 benign, 48 malignant, and 80 test) with resolutions around $775 \times 522$ or $589 \times 453$. **CRAG** Graham et al. (2019) targets gland segmentation in colon histopathology, comprising 213 annotated images, mostly sized $1512 \times 1516$. It is particularly challenging due to substantial variations in gland morphology and staining. Following prior works, 20% of the images were used as a fixed test set, and the remaining 80% were split into five folds for cross-validation on both datasets.

## 3.2 Implementation Details

All experiments were conducted on a single NVIDIA RTX 3060 GPU (16 GB VRAM) using Python 3.11, PyTorch 2.5, and CUDA 12.2. We adopted DeepLabV3+ with a ResNet-101 backbone as the segmentation model for both GlaS and CRAG datasets, training for 80 and 120 epochs respectively, with all images resized to $256 \times 256$.

To enhance generalization, we applied shared augmentations across branches, including random flips and rotations. Each student also received branch-specific augmentations tailored to its objective. The Color Student used either color jittering or histogram matching Ferrero et al. (2024) to improve robustness to appearance variations while preserving structure. In contrast, the Structure Student was augmented with elastic deformation Faryna et al. (2021) to better capture structural distortions and fine-grained morphology.

Models were optimized using AdamW (lr $= 1 \times 10^{-4}$, $\beta = (0.9, 0.999)$, $\epsilon = 1 \times 10^{-8}$, weight decay $= 0.05$) with a batch size of 4. During inference, the student model with the best validation performance was used for prediction.

## 3.3 Comparison with State-of-the-arts

| Labeled Ratio | Method | GlaS | | CRAG | |
|---|---|---|---|---|---|
| | | Dice ↑ | Jaccard ↑ | Dice ↑ | Jaccard ↑ |
| 100% | Fully-Supervised | **90.84 ± 0.28** | **83.72 ± 0.42** | **86.36 ± 0.73** | **77.38 ± 1.05** |
| 5% | Supervised Baseline | 72.31 ± 3.19 | 58.82 ± 3.43 | 63.87 ± 3.29 | 50.73 ± 2.78 |
| | Mean Teacher (NeurIPS 2017) | 77.23 ± 3.08 | 64.59 ± 3.34 | 71.71 ± 2.06 | 58.78 ± 2.35 |
| | UAMT (MICCAI 2019) | 79.43 ± 3.28 | 67.85 ± 3.75 | 72.01 ± 2.45 | 59.53 ± 2.30 |
| | Fixmatch (NeurIPS 2020) | 79.15 ± 1.75 | 67.66 ± 2.21 | 71.67 ± 3.36 | 58.83 ± 3.56 |
| | CPS (CVPR 2021) | 79.55 ± 2.01 | 67.84 ± 3.21 | 74.34 ± 2.19 | 61.36 ± 2.58 |
| | CT (MIDL 2022) | 79.86 ± 1.49 | 68.13 ± 3.64 | 71.84 ± 3.93 | 58.34 ± 4.79 |
| | CCVC (CVPR 2023) | 80.84 ± 1.75 | 68.88 ± 2.28 | 73.28 ± 1.87 | 60.48 ± 2.17 |
| | CorrMatch (CVPR 2024) | 79.85 ± 1.96 | 67.79 ± 2.79 | 69.08 ± 2.31 | 55.39 ± 2.88 |
| | FDCL (AAAI-25 Bridge Program) | 81.64 ± 1.08 | 70.15 ± 1.52 | 74.55 ± 1.51 | 61.93 ± 1.84 |
| | **CSDS (Ours)** | **82.86 ± 1.24** | **72.01 ± 1.81** | **75, 25 ± 1, 54** | **63, 00 ± 1, 67** |
| 10% | Supervised Baseline | 75.64 ± 3.99 | 62.48 ± 4.11 | 71.80 ± 3.11 | 58.91 ± 3.20 |
| | Mean Teacher (NeurIPS 2017) | 79.68 ± 3.33 | 68.08 ± 4.11 | 75.36 ± 2.17 | 62.77 ± 2.61 |
| | UAMT (MICCAI 2019) | 84.05 ± 0.78 | 74.20 ± 1.65 | 75.48 ± 3.16 | 63.05 ± 3.42 |
| | Fixmatch (NeurIPS 2020) | 81.13 ± 2.05 | 69.40 ± 3.71 | 74.87 ± 3.20 | 62.30 ± 3.38 |
| | CPS (CVPR 2021) | 84.29 ± 0.44 | 73.89 ± 0.88 | 78.08 ± 1.32 | 66.11 ± 1.79 |
| | CT (MIDL 2022) | 82.67 ± 1.15 | 71.65 ± 2.98 | 73.68 ± 1.86 | 60.44 ± 2.19 |
| | CCVC (CVPR 2023) | 83.78 ± 2.31 | 73.52 ± 2.99 | 74.97 ± 1.40 | 62.30 ± 1.76 |
| | CorrMatch (CVPR 2024) | 83.27 ± 2.23 | 72.59 ± 2.01 | 74.90 ± 1.41 | 61.93 ± 2.35 |
| | FDCL (AAAI-25 Bridge Program) | 84.35 ± 1.12 | 74.45 ± 1.14 | 76.28 ± 2.29 | 63.92 ± 2.88 |
| | **CSDS (Ours)** | **85.06 ± 0.57** | **74.87 ± 0.82** | **79.50 ± 0.88** | **68.14 ± 1.18** |

Table 1: Quantitative results on GlaS-2017 and CRAG-2019 datasets under two labeled ratios. We highlight the best performance for each metric in bold, and the second-best in underline.

We benchmark the proposed method against eight SOTA semi-supervised segmentation methods, covering both general-purpose and histopathology-specific approaches Tarvainen and Valpola (2017); Yu et al. (2019); Sohn et al. (2020); Chen et al. (2021); Luo et al. (2022); Wang et al. (2023); Sun et al. (2024); Nguyen et al. (2025). We report the average performance across five folds along with standard deviation.

### 3.4 Quantitative Results

As shown in Table 1, CSDS consistently outperforms previous SOTA methods under limited supervision on both GlaS and CRAG datasets. On the Glas dataset, with only 10% labeled data, CSDS achieves a Dice score of $82.86 \pm 1.24$ and a Jaccard index of $72.01 \pm 1.81$, which surpasses strong co-training baselines like FDCL and CPS. In 5% labeled ratio, CSDS narrows the Dice gap to full supervision to just 0.81%, while significantly outperforming teacher-student methods (e.g., Mean Teacher) and self-training methods (e.g., FixMatch). CSDS generalizes well across datasets, as evidenced by consistently outperforming all compared methods under all settings on CRAG, which presents more irregular and challenging nuclei structures.

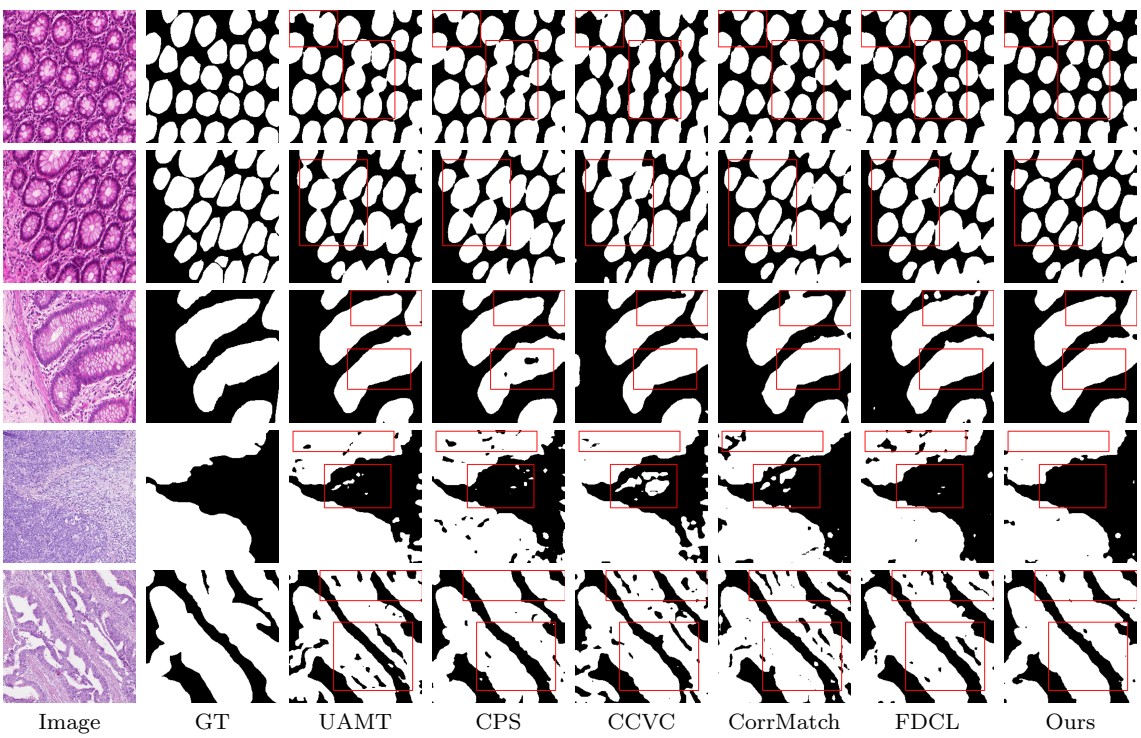

Image      GT      UAMT      CPS      CCVC      CorrMatch      FDCL      Ours

Figure 2: Qualitative results for different semi-supervised methods on 10% labeled data from two datasets. Rows 1 to 3 correspond to GlaS, while Rows 4 to 5 correspond to CRAG. The red boxes emphasize the differences in the results.

### 3.5 Qualitative Results

Figure 2 shows qualitative results on GlaS (Rows 1–3) with 10% labeled data. While most methods capture overall morphology, earlier approaches like UAMT and CPS struggle with boundary precision and separating clustered glands. FDCL and CorrMatch improve but still show artifacts. In contrast, CSDS produces cleaner, more accurate masks with sharper boundaries, aligning with the quantitative results in Table 1. Rows 4 and 5 show results on CRAG. While most methods fail to capture complete nuclei contours, causing under-segmentation or fragmented boundaries, CSDS achieves the most consistent results with clearer boundaries and better shape preservation.

### 3.6 Ablation study

**Effect of Model Components.** As shown in Table 2, removing either the Color or Structure Student leads to clear performance drops. Using both together significantly improves results, confirming their complementary roles in enhancing representation learning and segmentation.

| SupRatio | Teacher | Color Student | Structure Student | Dice (%) | Jaccard (%) |
|---|---|---|---|---|---|
| | ✓ | ✓ | – | $81.10 \pm 2.32$ | $69.53 \pm 2.97$ |
| 5% | ✓ | – | ✓ | $80.35 \pm 3.47$ | $68.75 \pm 4.11$ |
| | ✓ | ✓ | ✓ | $\mathbf{82.86 \pm 1.24}$ | $\mathbf{72.01 \pm 1.81}$ |
| | ✓ | ✓ | – | $84.37 \pm 1.27$ | $74.02 \pm 1.70$ |
| 10% | ✓ | – | ✓ | $84.07 \pm 1.55$ | $73.75 \pm 2.03$ |
| | ✓ | ✓ | ✓ | $\mathbf{85.06 \pm 0.57}$ | $\mathbf{74.87 \pm 0.82}$ |

Table 2: Ablation study using different Teacher, Color, Structure Student combinations.

**Effect of EMA Strategies.** As shown in Table 3, the *Mean* strategy, which updates the teacher model by averaging the parameters of both students, consistently achieves the best segmentation performance. In contrast, *Alternate*, which updates the teacher by switching between the two branches each epoch, and *Best student only*, which updates using the student with higher validation performance, perform worse. These results suggest that aggregating knowledge from both students yields more stable and informative teacher updates by reducing potential variance and bias introduced by a single student branch.

| SupRatio | EMA Strategy | Dice (%) | Jaccard (%) |
|---|---|---|---|
| | Alternate | $81.80 \pm 2.38$ | $70.44 \pm 3.19$ |
| 0.05 | Best student only | $81.60 \pm 1.88$ | $70.13 \pm 2.56$ |
| | Mean | $\mathbf{82.86 \pm 1.24}$ | $\mathbf{72.01 \pm 1.81}$ |
| | Alternate | $84.56 \pm 1.40$ | $74.30 \pm 1.93$ |
| 0.10 | Best student only | $84.34 \pm 1.64$ | $74.03 \pm 2.28$ |
| | Mean | $\mathbf{85.06 \pm 0.57}$ | $\mathbf{74.87 \pm 0.82}$ |

Table 3: Ablation study of different EMA strategies under varying supervision ratios.

## 4 Conclusion

In this work, we introduced Color-Structure Dual-Student (CSDS), a novel semi-supervised segmentation framework tailored for histopathology images. By decoupling color and structural cues into two specialized student networks guided by a shared teacher, CSDS captures domain-specific variations more effectively. An uncertainty-aware training strategy further improves pseudo-label reliability via color- and structure-adaptive weighting. Experiments on GlaS and CRAG under limited-label settings show that CSDS consistently outperforms strong baselines and recent state-of-the-art methods, highlighting the benefit of explicitly modeling histopathology-specific visual cues.

## Acknowledgments and Disclosure of Funding

This research was funded by NAFOSTED, Vietnam National Foundation for Science and Technology Development, grant number IZVSZ2_229539.

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
