# OpenReview forum: "Learning Disentangled Stain and Structural Representations for Semi-Supervised Histopathology Segmentation"
_MICCAI.org/2025/Workshop/COMPAYL — COMPAYL 2025_

### Official Review · Reviewer_5QBD · 2025-07-14
**Interesting Attempt at Disentangling Color and Structure in WSIs With a Reproducible Implementation but Insufficient Statistical Evidence for Claimed Dual-Student Architecture Benefits**

**Rating:** 2
**Confidence:** 3

**Review:**

Contribution:
This paper proposes a new self-supervised segmentation framework named Color-Structure Dual-Student (CSDS). The goal is to decouple staining and structural variations by training a dual-student and teacher network using different image augmentations for each student, as well as applying domain-specific prior in regard to color and structure to enhance the reliability of the pseudo-label generation. The authors claim that using this specific architecture, and thus modeling visual cues specific to histopathology, outperforms the state-of-the-art on the GlaS and CRAG datasets. However, the claims of significant performance gains seem overstated and from the current results it is not clear that the marginal performance gains are solely due to the model architecture presented in this paper.

Strengths:
- The overall presentation and writing of this paper is of hight quality.
- The authors acknowledge the variations in staining and structures which makes segmentation often a challenging problem, especially in gland segmentation. This is a relevant topic as this can hamper generalizability of deep learning models in general.
- The authors use generally available datasets and methods, and have published their code, which allows for reproducibility.
- The provided results are clear and the ablation studies on the model components and EMA strategies are also highly relevant.

Weaknesses:
- The paper's primary weakness lies in overstating performance improvements despite ambiguous statistical evidence. While the authors report confidence intervals, frequent overlap between proposed and baseline methods undermines claims of significant advancement. This issue becomes particularly evident in ablation studies, where confidence intervals often completely overlap, as demonstrated by the 10% labeled data scenario where "Color only" achieves 84.37 ± 1.27 [83.10, 85.64] compared to "Both students" at 85.06 ± 0.57 [84.49, 85.63]. Such overlap contradicts assertions of "significant" or "clear" performance gains. Nevertheless, the proposed method does exhibit reduced variance alongside marginally higher mean performance, suggesting modest improvements in prediction consistency rather than substantial performance breakthroughs.
- The ablation results, characterized by overlapping confidence intervals, cast doubt on the dual-student architecture as the primary factor behind marginal performance gains. Since individual students perform comparably to the dual-student setup, the claimed disentanglement of staining and structural variations appears questionable. The EMA weight averaging process may undermine student specialization by forcing both networks to learn from identical pseudo-labels while averaging their parameters. This mechanism could drive convergence toward similar learned representations, potentially negating any benefits of the dual-student approach compared to single-student training with combined augmentation strategies.
- The authors' quantitative analysis lacks information about whether image augmentations were applied when training baseline models. This omission is significant because augmentations matching those used in the proposed two-stream method could independently affect performance, separate from architectural improvements.

---

### Official Review · Reviewer_sks8 · 2025-07-15
**learning color and structure for gland segmentation**

**Rating:** 4
**Confidence:** 4

**Review:**

Summary:
- The authors propose a semi-supervised model for gland segmentation, designed to learn disentangled representations of staining and tissue structure, with a teacher-student framework.

Strengths
- very well-written paper
- good ablation studies and extensive comparison experiments using two datasets

Weaknesses
- Would other labeled ratios increase performance?
- overlapping performance intervals (across metrics and datasets) in comparison with other frameworks/ablation experiments, which should be assessed a bit more carefully.